# Effects of Task-Specific Training after Cognitive Sensorimotor Exercise on Proprioception, Spasticity, and Gait Speed in Stroke Patients: A Randomized Controlled Study

**DOI:** 10.3390/medicina57101098

**Published:** 2021-10-13

**Authors:** Kyung-Hun Kim, Sang-Hun Jang

**Affiliations:** 1Department of Physical Therapy, Gimcheon University, 214 Daehak-ro, Gimcheon-si 39528, Korea; huni040@naver.com; 2Department of Physical Therapy, Korea National University of Transportation, 61 Daehak-ro, Jeungpyeong-gun 27909, Korea

**Keywords:** cognitive sensorimotor exercise, task-specific, proprioception, spasticity, stroke

## Abstract

*Background and objectives*: Common problems in stroke patients include loss of proprioception, spasticity, and impaired gait. The purpose of this study was to examine the effects of task-specific training (TST) combined with cognitive sensorimotor exercise (CSE) on proprioception, spasticity and gait speed in stroke patients. *Materials and**Methods:* Thirty-seven subjects were randomly divided into three groups; (1) the TST after CSE group (Experimental I, *n* = 13); (2) the TST group (Experimental II, *n* = 12), and (3) a conventional physical therapy training group (control group, *n* = 12). Evaluations were performed before the commencement of training and again eight weeks after training was initiated. An electrogoniometer was used to evaluate proprioception variation. The composite spasticity score (CSS) and MyotonePRO were used to evaluate spasticity. In addition, 10 m walk test was used to assess gait speed. *Results*: After training, the Experimental I group showed significant improvement in proprioception compared to the Experimental II and control group (*p* < 0.05). In CSS, gastrocnemius muscle tone (GMT) and gait speed among three groups, Experimental I group differed significantly after eight weeks of training compared to the control group (*p* < 0.05). *Conclusions*: These findings suggest that the TST combined with CSE provided significant improvements in proprioception, spasticity, and gait speed.

## 1. Introduction

Stroke patients experience gait disorders due to various causes such as sensory impairment, spasticity, and motor impairment [1,2]. One of the important indicators for the functional outcome of stroke patients is independent gait ability. Thus, the main goal of rehabilitation is to reduce spasticity and gait ability in stroke patients [3].

Task-specific training (TST) is a widely known treatment approach focused on function in rehabilitation and used as a rehabilitation approach for most stroke patients [4,5]. Several studies showed that intervention with the TST was used to reduce gastrocnemius and soleus muscle spasticity and improve gait ability in stroke patients [6,7,8].

Looking at previous studies, various whole body vibration [9], percutaneous electrical nerve stimulation [10], cognitive behavioral therapy [11], eccentric strength exercise [12], repetitive transcranial magnetic stimulation [13], and robot therapy [14] and further studies have been conducted on treatment methods that combine TST. However, an objective effect on proprioception and functional ability has not been demonstrated. In prior combined methods, the first widely proposed cognitive sensorimotor training was Perfetti’s method [15,16]. Perfetti’s method is characterized by its focus on sensory retraining and particular joint location awareness [17]. Thus, cognitive sensorimotor exercise (CSE) is a special comprehensive rehabilitation program for retraining sensory-induced motor control and cooperative systematic guidance. Recently, a study on functional improvement reported that motor function was closely related to cognitive function [18]. CSE based on the learning theory is considered a method of learning from a pathological condition in motor function recovery that encourages patients to activate the cognitive process and causes a broad recovery from damage [15].

Despite a few studies on the effectiveness of CSE combined with the TST, more verification is needed to improve patient’s functional aspects [15,16,17,18,19]. Additionally, two recent systematic reviews demonstrated that TST was superior to Bobath therapy in improving gait and lower limb function [20,21]. To improve the proprioception, spasticity, and gait speed of stoke patients, the TST combined with CSE might be useful. Therefore, this study was conducted to determine the effect of the combined use of the TST after CSE on proprioception, spasticity, and gait speed in stroke patients. This study hypothesized TST combined with CSE (Experimental I group) would improve the proprioception, spasticity, and gait speed of stroke patients compared to Experimental II and control groups.

## 2. Materials and Methods

### 2.1. Participants

This study was an assessor-blinded, randomized controlled clinical design. This study was designed for 45 stroke patients who were in Bundang Jesaeng hospital located in Gyeonggi-do. We explained the study requirements and purpose in detail and collected written informed consent after reminding the patients that they had the right to withdraw from the study at any time. The study adhered to the Helsinki Declaration principles and received approval from the Sahmyook University Institutional Review Board (2-1040781-AD-N-01-2016009HR, 26 APRIL 2016). The trial was registered under trial registration no. KCT0006579.

The inclusion criteria were as follows: (1) diagnosis of stroke of onset had passed 6 months of more months; (2) ability to walk at least 10 m; (3) mini-mental state examination-Korean (MMSE-K) scores of 24 or higher; (4) ability to communicate and follow the instructions; (5) Brunnstrom exercise recovery stages of 3–4 or higher; (6) no problem walking due to ankle joint contractures; (7) no sensory deficiencies of the lower extremities; (8) and voluntarily provided informed consent prior to participating were included in this study.

The exclusion criteria were as follows: (1) vestibular problems; (2) cerebellar-related diseases; (3) visual or hearing impairments; (4) could not readily participate due to severe cognitive decline or aphasia, and (5) unilateral neglect. 

### 2.2. Sample Size Calculation

The sample size estimation was based on data collected from a pilot study (10 stroke patients). We used G*power 3.1.9.4 software (Heinrich-Heine-University Düsseldorf, version 3.1.9.4, Düsseldorf, Germany) to calculate the sample size [22]. The effect size variable was the proprioception error. The input parameters were the number of groups (3), SD (0.98), group 1 (mean: 3.21, size: 4), group 2 (mean: 1.72, size: 3), group 3 (mean: 0.58, size: 3). Thus, a total of 36 study subjects were calculated, 12 in each group, where the alpha error was 0.05, power was 0.80, and effect size (f) was 1.1240267. Thus, 55 participants were recruited in consideration of drop-out.

### 2.3. Test Design

The participants were randomly divided into three groups: (1) the TST combined with CSE group (Experimental I group); (2) TST group (Experimental II group), and (3) conventional physical therapy group (Control group). Block randomization was determined using a randomization procedure in which each participant drew a ball from a box containing three balls with markings 1 (experimental I group), 2 (experimental II group), or 3 (control group). The interventions were conducted over eight weeks and evaluations were performed one week before and after training, before the experiment, and at the eight weeks. The evaluation examined proprioception error, the composite spasticity score, gastrocnemius muscle tone, and the 10 m walk test. Depending upon the subject’s exercise capacity, the training was stopped if they could not maintain training for 30 min, while a 5 min rest period was allowed if they showed fatigue, reported pain, demonstrated abnormal breathing, or appeared flushed [23]. With the exception of this training, routine therapy was allowed in the three groups.

### 2.4. Intervention

#### 2.4.1. Experimental I Group

The Experimental I group performed sense, trunk stability, lower-extremity movements, and gait training in a sitting or standing position by applying CSE and the TST. In this study, 30 min of CSE and 30 min of the TST were performed five times per week for eight weeks. Two physical therapists performed treatment to reduce treatment bias. One physical therapist with more than five years of clinical experience performed the CSE exercises. The TST training was conducted by the researcher who has more than 5 years of clinical experience.

The participants were trained to recognize their bodies using visual and somatosensory techniques. The CSE consisted of proprioceptive training, tactile training, heel pressure, and space tasks. For CSE using a spatial task, exercise programs were given to distinguish distance and direction. The CSE was applied for a total of 30 min for 5 min each with the subjects seated. After modifying and complementing the program using suggestions based on the studies of Chanubol et al. (2012) and Cappellino et al. (2012), we performed the experiment with advice from senior physical therapists and rehabilitation doctors [17,24] (Figure 1) (Table 1).

The 30 min TST consisted of the following: 10 min of trunk stability and sit to stand training while controlling the movement trunk and lower extremities [25], 10 min of lower extremity movements and gait training while controlling scapular movement [26,27], 10 min of progressive body weight support treadmill training while controlling the movement of the lower extremities [28].

#### 2.4.2. Experimental II Group

The Experimental II group performed trunk stability and sit-to-stand training while controlling the movement of the trunk and lower extremities, lower extremity movements and gait training while controlling scapular movement, and progressive body weight support treadmill training while controlling the movement of the lower extremities. Experimental II group received 30 min of the TST and 30 min of conventional physical therapy five times per week for eight weeks.

#### 2.4.3. Control Group

The conservative physical therapy program consisted of ROM exercises, stretching exercises, upper- and lower-extremity muscle strengthening exercise, ground gait training, bike exercises, balance training, and superdynamics exercises. The control group received conservative physical 30 min twice daily, five times per week for 8 weeks.

### 2.5. Evaluation

Three evaluators were assigned to conduct the evaluation. Three physical therapists with a master’s degree and more than five years of clinical experience performed the measurements. Evaluators blinded to group allocation performed the evaluations.

#### 2.5.1. Proprioception

The proprioception test for the present study was conducted using an electrogoniometer. Proprioception involved specified angles (30°, 60°, 90°, 120°, 150°) and a 60 cm × 60 cm × 1 acrylic assessment board [29,30]. The subjects wore an eye patch to block visual information while sitting on a chair. A target angle is randomly selected from among the five given angles. The subject was asked to flexion the knee to the target angle and then return to the initial position. Again, the patient themselves was instructed to flexion the knee to the target angle. Finally, the difference in degree was measured using an electrogoniometer. The intra-rater reliability was high at r = 0.86~0.87 for the sitting straight position [31].

#### 2.5.2. Spasticity

Composite spasticity score (CSS) was used to assess the stiffness of the planter flexor [32]. CSS comprises 3 measure: (1) a 5-point scale to Achilles tendon jerk; (2) an 8-point to resistance to passive stretch of ankle dorsiflexion; (3) a 4-point to from ankle clonus. A score of 0–9 means mild spasticity; a core of 10–12 means moderate spasticity; a score 13–16 means severe spasticity [33,34].

#### 2.5.3. MyotonPRO

To evaluate gastrocnemius muscle tone, a MyotonPRO (Myoton AS, Talinn, Estonia) was used. For the measurement method, if a force of 0.18 N (pre-load force) is applied to the skin, the equipment applies an impulse of 0.58 N to the skin, the color changes from red to yellow-green, and an impulse maintained for approximately 5 s is recorded as usable data. Frequency indicated the muscle ability to resist external force of passive stretching. The higher the value the greater the muscle tone [35,36].

#### 2.5.4. 10 m Walk Test

The 10 m walk test was a simple method of predicting the gait speed of stroke patients. In this method, patients were asked to stand behind the start line and to walk, using a walking aid if necessary, and at their referred speed, until they crossed the 5 meter line, turned and walked back until they crossed the start line again. A 10 m walk was measured using a stopwatch, omitting the time taken to turn. This 10 m walk test and inter-rater reliability was high (r = 0.87) [37].

### 2.6. Data Analysis

PASW 20.0 for Windows was used for the data analyses. The variables of gender, diagnosis, affected side, and Brunnstrom recovery stage were assessed with the Chi-squared test, while height, weight, age, MMSE-K, K-NIHSS, onset year, and homogeneity of the dependent variables before training of the three groups were assessed using one-way analysis of variance (ANOVA). A normality test was performed using the Shapiro–Wilk test. For the difference variable among the three groups before and after the training, 2-way ANOVA was used: the group-by-time (between-factors) was used for changes during each time period according to the experimental conditions. If there was significant group-by-time, post hoc analysis Bonferroni test was used. The statistical significance of the data was accepted at values of α = 0.05.

## 3. Results 

### 3.1. General Characteristics of Participants

Table 2 indicated the general characteristics of participants. Fifty-five research subjects were selected for the study, but 10 did not meet inclusion criteria. During the 8-week experimental period, two, three, and three participants dropped out from Experimental I (*n* = 13), Experimental II (*n* = 12), and the control group (*n* = 12), respectively, for a final total of 37 study participants (Figure 2).

### 3.2. Comparison of Proprioception Error between the Three Groups

There was a significant difference in the proprioception error after eight weeks of training among the three groups. In particular, post-hoc testing revealed that the changes in the proprioception error variable Experimental I group (mean change −3.10 ± 1.30) were significantly decreased from those in Experimental II (mean change −1.74 ± 1.06) and Control group (mean change −0.58 ± 0.35) (*p* < 0.05). Additionally, Experimental II showed a significant decrease compared to the control group (Table 3).

### 3.3. Comparison of CSS and GMT among the Three Groups

There was a significant difference in the CSS and gastrocnemius muscle tone (GMT) after eight weeks of training among the three groups. In particular, post-hoc testing revealed that the changes in the CSS and GMT variables in Experimental I (mean change −1.54 ± 0.78, −0.77 ± 0.41, respectively) and Experimental II (mean change −1.00 ± 0.74, −0.73 ± 0.38, respectively) groups were significantly decreased from those of the Control group (mean change −0.25 ± 0.45, −0.28 ± 0.13, respectively) (*p* < 0.05) (Table 4).

### 3.4. Comparison of 10MWT among the Three Groups

There was a significant difference in the 10MWT after eight weeks of training in among the three groups. In particular, post-hoc testing revealed that the changes in the 10MWT variables in Experimental I (mean change 0.31 ± 0.08) and Experimental II group (mean change 0.25 ± 0.09) were significantly increased from those of the Control group (mean change 0.10 ± 0.06) (*p* < 0.05) (Table 5).

## 4. Discussion

This study was conducted to evaluate the effect of the TST after CSE on proprioception, spasticity, and gait speed in stroke patients. The TST and CSE were conducted for eight weeks and proprioception variation, spasticity, and gait speed of the stroke patients were analyzed. The main findings of this study are as follows: (1) Experimental I group showed greater significant improvement in proprioception than two groups; (2) Experimental I and Experimental II and groups showed greater significant improvement in CSS and gait speed than the control group.

The TST and CSE focused on the ICF activity level. Therefore, given that the treatment provided here was a combination of the two approaches, the ICF theory is significant since it uses both body function, and structural and activity levels. 

The proprioception error of the Experimental I group showed significant differences after eight weeks of training compared to the Experimental II group and the control group. In 2014, Jung et al. reported that weight shifting training improved trunk control and proprioception in stroke patients [38]. In 2019, Lim reported that a multisensory training program significantly improved proprioception and balance ability in stroke patients [30]. The application of the TST after CSE changes the sensory input to the joint and muscle receptors as well as skin receptors of the sole [39]. Proprioception was repeatedly stimulated in the present study, and the performance of exercise tasks is believed to have improved functional recovery. This is considered the result of implementing a program in which the stimulation of proprioception applied more neurological weight than that of the other senses.

Many stroke patient survivors develop spasticity. Spasticity is usually abnormal association movements, in particular, the deformity of the ankle joint is accompanied by paralysis. The CSS of the Experimental I group showed significant differences after eight weeks of training compared to the control group. In 2016, Lee et al. reported that the whole body vibration plus task-related training on the excitability alpha motor neurons decreased the modified Ashworth spasticity scores in stroke patients [9]. Na et al. (2008), reported the Transcutaneous electrical nerve stimulation combined with task-related training showed more statistically significant differences than the placebo + task-specific training (TRT) group in decreased platarflexor spasticity, improved dorsiflexor and palntarflexor strength [10]. In another study, Wu et al. (2006) applied repetitive manual ankle joint extension movement in 12 chronic stroke patients with spasticity of the medial gastrocnemius to determine its effect on walking and spasticity [40]. It seems like rigidity was reduced because the constant sensory input of the lower limbs and the TST improved the proprioception of the subjects, thereby stimulating the responses of the gastrocnemius muscle on the paretic side. It is believed that the rigidity of the subjects decreased after increasing the movement of the lower limbs, trunk control by the TST, and the weight applied during treadmill gait training by means of CSE.

The 10MWT of the Experimental I group showed significant differences after eight weeks of training compared to the control group. In 2015, Jeon et al. reported that the task oriented training interventions showed a significant increase in gait velocity, gait endurance, and muscle strength of the lower extremity after the experiment [41]. Lin (2006) reported that changes in ankle joint position sense produced significant differences in walking velocity and stride length in a study on the effect of proprioceptive sense and motor function on the walking of stroke patients [42]. CSE and TST patterns to improve stability and mobility of the hip, knee, and ankle joints, properly adjusted the selective movements of the trunk and lower-extremity muscles in a state in which trunk adjustment stability was improved through trunk movement. Therefore, it is thought that the selective movement was activated, and the walking ability of stroke patients was improved.

This study had limitations. First, it may be difficult to expect the same results for acute and subacute patients because this study targeted chronic stroke patients an average of six months post-stroke. In acute stage patients before 6 months, neuroplasticity occurs actively. The purpose of this study is to investigate the effects of intervention methods on stroke patients who have passed 6 months from the onset. Second, the daily activity level of rehabilitation hospitalized post-stroke patients was not controlled. Third, this study did not conduct follow-up tests on the participants. Additionally, it was not confirmed how the training method affected the balance and gait parameters. Therefore, studies that address the above limitations, as well as various types of studies scientifically proving that rehabilitation programs improve the proprioception, spasticity and gait speed of stroke patients are required in the future.

## 5. Conclusions

This study found that the TST after CSE was effective in improving proprioception, spasticity and gait speed in stroke patients. Our results indicate that CSE and TST training can be considered as a potential method to improve the proprioception, spasticity and gait speed in stroke patients. Diversified CSE will need to be developed for broader application of the combined approach as a therapeutic intervention for the functional recovery of stroke patients.

## Figures and Tables

**Figure 1 medicina-57-01098-f001:**
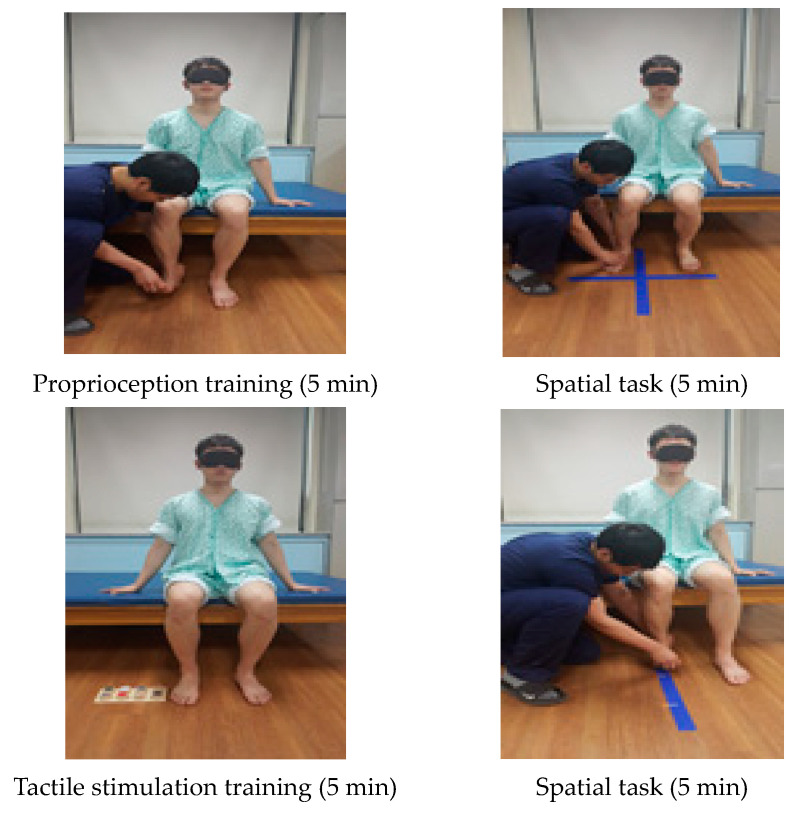
Cognitive sensory exercise.

**Figure 2 medicina-57-01098-f002:**
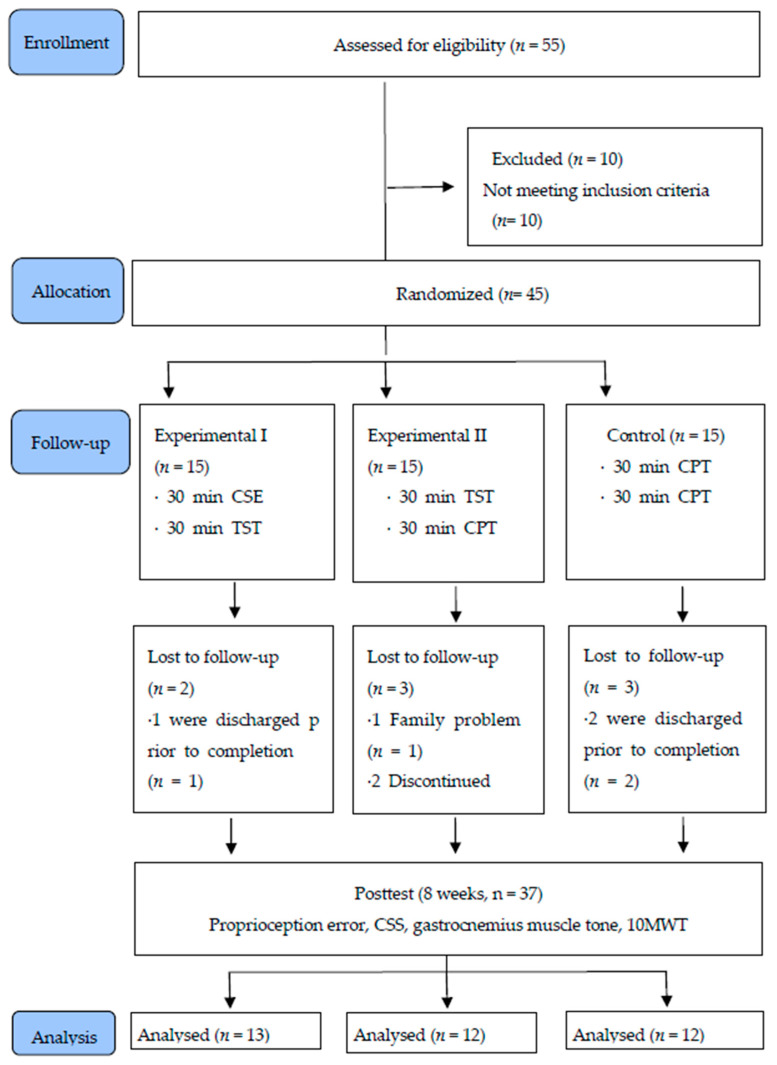
Flow diagram of the study.

**Table 1 medicina-57-01098-t001:** Cognitive sensorimotor exercise program.

Cognitive Sensorimotor Exercise Program	
Proprioception training (5 min)	The training was performed in the initial position and in the final position for proprioception after putting pressure on the ankle joint. The training was performed in the initial position and in the final position for position sense.
Tactile stimulation training (5 min)	CSE training using a tactile task was given to distinguish surface materials and friction sense. Subjects used visual and somatosensory techniques to differentiate each sense.
Pressure stimulation training (5 min)	Subjects presented a cognitive problem to distinguish the difference in the degree of the sponge pressure on the trunk in the sitting posture and asked the subjects to distinguish between the vision and somatosensory senses while conducting a task.
Spatial task (5 min)	A spatial task was given to distinguish where the patient’s foot was positioned in four areas. A line was drawn vertically on the floor straight from the patient’s affected knee and this line was horizontally divided in half.
Spatial task (5 min)	The patient’s heels were put together, and three lines were made depending on the knee angle. Training was performed to see if there were any changes in distance. If the patient was able to complete this task, a line was added.
Spatial task (5 min)	For the spatial cognition training, patients were instructed to trace an oval shape in front of them, and the position on the floor was expanded from a small one to bigger ones.

**Table 2 medicina-57-01098-t002:** General characteristics of participants.

	Experimental I (*n* = 13)	Experimental II (*n* = 12)	Control Group (*n* = 12)	*p*
Height (cm)	165.55 ± 7.51	164.93 ± 7.00	168.04 ± 4.70	0.474 ^a^
Weight (kg)	61.48 ± 9.17	65.83 ± 9.83	64.35 ± 9.55	0.512 ^a^
Age (year)	50.23 ± 14.89	52.75 ± 17.00	55.08 ± 10.55	0.704 ^a^
MMSE-K (score)	27.77 ± 1.64	27.17 ± 1.53	27.50 ± 1.09	0.586 ^a^
K-NIHSS (score)	9.61 ± 2.33	9.58 ± 2.43	9.75 ± 2.73	0.455 ^a^
Onset (months)	12.07 ± 3.57	13.17 ± 3.90	11.83 ± 3.71	0.649 ^a^
Gender (male/female)	7/6	7/5	8/4	0.805 ^b^
Diagnosis (infarction/hemorrhage)	6/7	6/6	8/4	0.556 ^b^
Affected side (Left/Right)	6/7	6/6	7/5	0.826 ^b^
Brunnstrom recovery stage (3/4)	8/5	8/4	6/6	0.695 ^b^

^a^ one-way ANOVA, ^b^ Chi-square test, MMSE-K, mini-mental state examination-Korea version, K-NIHSS, is the Korean national institute of health stroke scale.

**Table 3 medicina-57-01098-t003:** Comparison of proprioception error among the three groups.

	Experimental I (*n* = 13, A)	Experimental II (*n* = 12, B)	Control Group (*n* = 12, C)	df	ES	F(p)	Post-Hoc
Proprioception Error (Degree)							
Pretest	11.48 ± 1.58	11.84 ± 1.53	11.57 ± 1.57	2	0.0980	0.177 (0.839)	
Posttest	8.38 ± 1.33	10.10 ± 1.38	10.99 ± 1.46	2	0.7870		
change	−3.10 ± 1.30	−1.74 ± 1.06	−0.58 ± 0.35	2	1.1525	20.054 (0.000)	A > B > C
*t*(*p*)	8.606 (0.000)	5.717 (0.000)	5.619 (0.000)				

df: degree of freedom, ES: effect sizes f.

**Table 4 medicina-57-01098-t004:** Comparison of CSS and GMT among the three groups.

	Experimental I (*n* = 13, A)	Experimental II (*n* = 12, B)	Control Group (*n* = 12, C)	df	ES	F(p)	Post-Hoc
Composite Spasticity Score (Score)							
Pretest	11.23 ± 0.83	11.25 ± 0.87	11.17 ± 0.83	2	0.0400	0.032 (0.968)	
Posttest	9.69 ± 1.03	10.25 ± 0.75	10.92 ± 0.79	2	0.5876		
change	−1.54 ± 0.78	−1.00 ± 0.74	−0.25 ± 0.45	2	0.8046	11.433 (0.000)	A, B > C
*t*(*p*)	7.146 (0.000)	4.690 (0.001)	1.915 (0.082)				
Gastrocnemius Muscle Tone (Hz)							
Pretest	15.88 ± 1.96	15.18 ± 1.75	14.69 ± 1.50	2	0.2830	1.450 (0.249)	
Posttest	15.10 ± 2.10	14.44 ± 1.74	14.42 ± 1.54	2	0.1787		
change	−0.77 ± 0.41	−0.73 ± 0.38	−0.28 ± 0.13	2	0.7129	8.397 (0.001)	A, B > C
*t*(*p*)	6.774 (0.000)	6.720 (0.000)	7.545 (0.000)				

df: degree of freedom, ES: effect sizes f.

**Table 5 medicina-57-01098-t005:** Comparison of 10MWT among the three groups.

	Experimental I (*n* = 13, A)	Experimental II (*n* = 12, B)	Control Group (*n* = 12, C)	df	ES	F(p)	Post-Hoc
10 m Walk Test (m/s)							
Pretest	0.72 ± 0.22	0.72 ± 0.22	0.71 ± 0.21	2	0.0213	0.007 (0.993)	
Posttest	1.03 ± 0.23	0.97 ± 0.25	0.82 ± 0.26	2	0.3533		
change	0.31 ± 0.08	0.25 ± 0.09	0.10 ± 0.06	2	1.1041	22.194 (0.000)	A, B > C
*t*(*p*)	13.647 (0.000)	9.537 (0.000)	5.778 (0.000)				

df: degree of freedom, ES: effect sizes f.

## Data Availability

Not applicable.

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
