# Peer review of "Effects of Task-Specific Training after Cognitive Sensorimotor Exercise on Proprioception, Spasticity, and Gait Speed in Stroke Patients: A Randomized Controlled Study"

_medicina, 2021, doi:10.3390/medicina57101098_

Round 1
Reviewer 1 Report
Thank you for the opportunity to review this manuscript. The purpose of the study was to identify the effects of a combined task-specific and sensorimotor training on stroke patients motor abilities compared to an isolated TST and a standard therapeutical protocol.
I completely see the interest of the topic for the readership of the scope of medicina, however, I’ve some comments that need to be addressed before considering the paper for publication.
Generally, I encourage you to let your manuscript be revised by a English native speaker.
Abstract:
The Abstract is generally well written. However, thoroughly revise the structure. For example, the Abstract lacks of a clearly outlined Results section.
Use different wording than subjects. This is pretty unusual at present, better use participants or just individuals that participated in your study.
Manuscript:
Introduction:
General comments:
Generally, the introduction covers all necessary content deduce the purpose of the study. However, there are still some modifications that you should consider to revise.
L27-28: This phrase needs referencing.
L31-32: This sentence needs revision for a better and more logic understanding.
L35-39: This sentence is quite long. Disentangle the latter part from the first part and give references for their prove of the potential effects.
L45-48: Could you please enlarge the scope and explanation of CSE based training.
Please provide a research hypothesis according to your purpose.
Methods:
L58: Grammatically revise.
L59: Give full details of the affiliated institution.
L65-71: Please list the inclusion criteria grammatically correct.
L72-75: Same as for inclusion criteria.
L76 et seq.: Well elaborated section on the sample recruitment criteria.
L84: Is procedure a well-fitting term here? Test Design?
L90-92: Generally, in intervention studies it is quite normal to do an additional testing after a retention interval, 3 or 6 months after the last intervention to get a result about the sustainability of the intervention. Why didn’t you include such a retention interval?
Section 2.4.1, 2.4.2, and Figure 1: Please give more detailed information of the tasks that were used in the intervention program. With one picture and the caption proprioception training, I do not actually know what kind of exercises were conducted. This is mandator for all tasks you included in the intervention program (i.e., spatial task, tactile stimulation training and so on).
L141-143: Revise sentence.
Results:
L179: Grammatically revise sentence.
L179 et seq.: Refuse to use the term subject in a manuscript.
L179-180: Grammatically revise sentence.
Table 1: Give ranges of minima and maxima for the MMSE-K and K-NIHSS scores. Also, you need the clarify thresholds of these scores, when a individuals is considered impaired or not.
Figure 2: I appreciate the work of Figure 2. Hence, it does not add new information you can consider to remove the Table.
L210-213: This sentence requires grammatical revision. And, how is the statement supported by the results? “Ex group I were decrease from those in Ex II”. However, Experimental I has the largest reduction of proprioception error compared to Experimental I and Control group. How is this to understand by a decrease??
L217 et seq.: I can you also not follow here. I do not understand why the Experimental groups are decreased in their CSS and GCMM compared to the control group as both groups improved their score.
You have to re-consider the presentation of your results: Are you interested in reporting the changes between the groups or the changes within the groups? You should align those results according to your hypothesis and research purpose.
Discussion:
L232 et seq.: Please give us your main finding in the first section of the Discussion.
L238-241: Sentence requires revision. Accordingly, I do not understand why your repetitively highlight the benefits of TST here. You did that already in the Introduction. How are your findings fit to the statement?
L242-243: What does that mean interpretation-wise?
L253-254: Revise sentence for understandability.
L253 et seq.: I do not fully see how your argumentation and interpretation is linked to the enrolled studies.
L269-275: So what? Discussion is not meant to just repeat a result and give any literature references that say the similar. In discussing your findings, you should give us logic interpretation of what the findings mean, in general, in specific and how that is linked or underlined by findings of literature. You should not only list your results again and list other results that fit kind of arbitrarily.
L276-277: This statement is in contrast to your sample size calculation with g-power. If doing a good power calculation, as you did, your sample should have an acceptable amount of participants to make a generalizability feasible.
L278-279: If you chose this peer, which is totally acceptable in stroke patients, your results and findings are firstly directly connected to your sample. So, place your interpretation on the sample instead of saying the group was not valid for acute and subacute patients with stroke. This was not the purpose of your study, right?
L279-281: If you mention the limitations, you should also give short explanations why this is limiting your study. Just mentioning is not sufficient to address the needs of a readership.
Conclusion:
L287-288: This is the first time that you deduced such a clear interpretation out of your results. After all what I’ve read, I did not expect that you can formulate such a common, general conclusion. If so, you have to re-arrange your Discussion, especially and give us a clearer background and analysis how this saying is supported by your results.
Reviewer 2 Report
Review
of the article
«Effects of Task-Specific Training after Cognitive Sensorimotor Exercise on Proprioception, Spasticity, and Gait Speed in Stroke Patients: a Randomized controlled study»
Authors:
Kyung-Hun Kim, Sang-Hun Jang
The research presented for review is devoted to examine the effects of Task-specific training (TST) combined with cognitive sensorimotor exercise (CSE) on proprioception, spasticity and gait speed in stroke patients. This study was conducted to determine the effect of the combined use of the TST after CSE on proprioception, spasticity, and gait speed in stroke patients.
The authors correctly identified the materials and research methods. The study adhered to the Helsinki Declaration principles and received approval from the Sahmyook University Institutional Review Board, which is confirmed by the relevant regulatory documents. For inclusion in the study, participants in the experiment were selected according to specific criteria: who had stroke of onset of 6 months prior; could walk at least 10 m; scored more than 24 points on the Mini Mental State Examination-Korean (MMSE-K); could communicate and follow the instructions, were in Brunnstrom exercise recovery stages of 3–4; had no problem walking due to ankle joint contractures; had no sensory deficiencies of the lower extremities; voluntarily provided informed consent prior to participating were included in this study.
To conduct a reliable experiment, the participants were randomly divided into three groups: experimental group 1 (2the TST combined with CSE group); experimental group 2 (TST group); control group (conventional physical therapy group). The interventions were conducted over eight weeks and evaluations were performed one week before and after training, before the experiment, and at the eight weeks.
Three evaluators were assigned to conduct the evaluation. Three physical therapists with a master’s degree and more than five years of clinical experience performed the measurements: proprioception, spasticity, myotonPRO, 10-m walk test. As a result of experimental work the proprioception error of the experimental group 1 showed significant differences after eight weeks of training compared to the experimental group 2 and the control group. The CSS of the experimental group 1 showed significant differences after eight weeks of training compared to the control group. The 10MWT of the experimental group 1 showed significant differences after eight weeks of training compared to the control group.
Although the study had limitations, this study found that the TST after CSE was effective in improving proprioception, pasticity and gait speed in stroke patients.
Author Response
Please see the attachment.
Thank you

This manuscript is a resubmission of an earlier submission. The following is a list of the peer review reports and author responses from that submission.